

# NON-LINEAR EFFECTS IN ELECTROMAGNETIC WAVE ACTIVITY OBSERVED IN THE RELEC EXPERIMENT ON-BOARD VERNOV MISSION

**Mikhail I. Panasyuk[1,2], Sergey I. Svertilov[1,2], Stanislav I. Klimov[3], Valery A. Grushin[3],**
**Denis I. Novikov[3], Yuri Ya. Ruzhin[4], Yuri M. Mikhailov[4], Csaba Ferencz[6], Peter Szegedi[7],**
**Valery E. Korepanov[5], Vitaly V. Bogomolov[1,2], Gali K. Garipov[1], Serhiy V. Belyayev[5],**
**Olexander N. Demidov[5]**

*1 – M.V. Lomonosov Moscow State University, D.V. Skobeltsyn Institute of Nuclear Physics*
*2 – M.V. Lomonosov Moscow State University, Physics Department*
*3 – Space Research Institute, Russian Academy of Science, Moscow, Russia*
*4 – N.V. Puchkov Institute of the Earth magnetism, Ionosphere and radio-wave propagation, Russian Academy of Science, Troitsk, Russia*
*5 – Institute of Space Research, Ukrainian Academy of Science and National Space Agency, Lviv, Ukraina*
*6 – Eötvös University, Space Research Group, Budapest, Hungary*
*7- BL Electronics Ltd., Solymár, Hungary*

The experiments on-board Vernov satellite were aimed on the study of high energy (relativistic and sub-relativistic) electron acceleration and losses in the trapped radiation areas as well as high altitude electric
discharges in the upper Atmosphere. A separate task was study electromagnetic-wave phenomena in the near Earth space and the upper Atmosphere. During observations on 10 December 2014 interesting phenomena were discovered. They are connected to non-linear effects in wave activity of the type of two or three wave decays as well as splitting into two wave structures. Whistlers with specific unusual temporal structure of swallowtail type were observed on the spectral diagrams (sonograms), which were
obtained for this time. It was shown that such signals can be caused by seismic activity. The signals of the type of whistler with long tail were also observed. Such signals were also detected by ground stations.

### 1. Ingtroduction.

The aim of scientific experiments with RELEC instruments on-board Vernov satellite (Panasyuk et al., 2016a) was study of the magnetospheric relativistic electron precipitation as well as transient, i.e. short-time sporadic phenomena in the Earth atmosphere. Such events can be caused by the physical processes of different nature ongoing with the release of large energy during short time intervals. In the Earth's atmosphere such phenomena include high-altitude
atmospheric discharges, including some types of thunderstorms, as well as Transient Luminous Events (sprites, elves and "blue jets") and their manifestations in different ranges of the electromagnetic spectrum.

The problem of acceleration and precipitation of energetic (with energy >100 keV) magnetospheric electrons is still not completely known in detail, especially in view of many
cases of frequent failures of satellite systems and payloads due to the impact of high fluxes of relativistic electrons. Precipitation is one of the main process of energetic electron losses in radiation belts. Precipitated electrons affect the properties of the upper Atmosphere, change the electrical and chemical properties of the stratosphere and the mesosphere, that leads, among other things, to the destruction of ozone (see, e.g. Thorne, 1977). The release of relativistic
electrons into the atmosphere increases strongly during a magnetic storm. To study the decrease of the trapped energetic electron fluxes during magnetic storm, it is necessary take into account adiabatic deceleration as a result of a decrease in the magnitude of the magnetic field at the equator, the change in the configuration of the drift trajectories, as a result of which the closed drift trajectories become open and radial diffusion. Kanekal et al., 2000 showed that the
precipitation into the loss cone obtains significant input from the decreasing relativistic electron fluxes comparing SAMPEX and Polar satellites observations. The precipitations in the loss cone occur as a result of a change in the pitch-angle distribution of the particles mainly due to wave-particle interaction. Calculation of the corresponding electron loss rates requires information



about the spectra of the excited waves. Several wave modes propagate and are generated in the magnetosphere, which can interact with relativistic electrons: They are whistlers, whistler-mode waves, electromagnetic ion cyclotron and electrostatic ion cyclotron waves. The waves are often generated by particles of lower energies, these waves then precipitate particles of higher energies (parasitic diffusion).

Whistler mode waves and electrostatic ion-cyclotron waves scatter most effectively electrons with energies <1 MeV, but they also can scatter particles with higher energies. Electromagnetic ion-cyclotron waves scatter electrons with energies >1 MeV more effectively. Precipitation in the loss cone is connected usually with gyro-resonance interaction of particles with whistler-mode waves, electromagnetic ion-cyclotron (EMIC) waves and plasmaspheric hisses. This problem was studied in details, see e.g. Kennel and Petscheck, 1966; Horne and Thorne, 2003; Thorne et al., 1977; Summers and Thorne, 2003; Albert, 2003; Meredith et al., 2006; Shprits et al., 2008 a,b and Ferencz et al., 2001.

ULF-ELF-VLF electromagnetic radiation plays the same role for the study of plasma processes in space as seismic waves for the Earth structure study. In comparison with electromagnetic processes in other environments, waves in plasma have a number of definite characteristic peculiarities. The resonance effect is the most important. It occurs due to the wave-particle interaction, wave transformations, resonator and waveguide formation. Due to the resonance effect, ultra-low frequency waves provide information about dynamic phenomena in the near-Earth space and the upper Atmosphere. By this they reach sufficiently large amplitudes to have significant influence on the plasma fluxes and effectively accelerate electrons in the magnetosphere. It was shown recently, that not only magnetosphere – ionosphere phenomena are accompanied by plasma disturbance, but also ground geophysical ones caused by large energy release such as explosions, hurricanes, thunderstorms and earthquakes.

Some natural electromagnetic phenomena known as space whether and occur in the solar wind – magnetosphere – ionosphere – Earth atmosphere system generate electromagnetic waves that can be detected in the ionosphere. Whistlers generated by thunderstorm discharges and detected by satellites are typical example of such emissions.

Atmospherics or spherics are electromagnetic signals produced by lightning discharges. The average occurrence of lightnings is about 100 strikes per second. A lightning discharge consists of 2 stages. There are pre-discharges and main discharges, which are differed by current and spectrum of emitted radio-waves. Ultra-long waves are emitted in the main discharge, while long, middle and even short waves are emitted in the pre-discharge.

The maximum of spheric energy lies in the range of about 4-8 kHz. If spherics are produced by nearby thunderstorms, their spectrum is determined only by the emission spectrum of thunderstorm discharge. If the source is a distant thunderstorm, its spectrum is determined also by the conditions of radio-wave propagation from the thunderstorm to the radio receiver. Spherics have a weak attenuation and can propagate over long distances.

Spherics may penetrate into the ionosphere and propagate along the magnetic field line, reaching the Earth-Ionosphere waveguide again in the other hemisphere and can be recorded on the ground. These waves exhibit frequency changes versus time and called whistlers. Their peculiarity is associated with Very Low Frequency wave propagation in magneto-ionic medium, see e.g. Hellivell, 1965; Ferencz et al. 2007, 2009; Ferencz et al., 2001.

The type of spheric spectrum is determined by magnetic field intensity and electron concentration along trajectory. The spectrum covers frequencies from hundreds Hz up to 20-30 kHz (Klimov et al., 2014). The spheric property analysis allows us to determine the electron concentration distribution along the propagation path, see e.g. Park, 1972; Lichtenberger, 2009. A rare phenomenon called knee whistlers (Carpenter, 1963) may be used to determine the location of the plasmapause Low-frequency branches of the spheric spectrum (ion whistlers) were detected on the satellites at frequencies below 400-500 Hz, over which the relative concentrations of ions and electrons, as well as other parameters of the ionosphere can be determined.



**2. Magnetic-wave instruments PSA – RFA as a part of the RELEC scientific payload on-board Vernov satellite.**

*2.1. RELEC payload.*

The RELEC scientific payload (Panasyuk et al., 2016 b,c) was operated on-board small spacecraft Vernov between July – December, 2014. It was named in honor of pioneer of space research in Soviet academician Sergey Nikolaevich Vernov. The spacecraft was elaborated and manufactured by Lavochkin space corporation and had the following main parameters:

    - mass – 283 kg;

- orientation accuracy – 6′;

    - stabilisation accuracy – $0.0015^{o}$/s;

    - data transmition rate – 5 Mbit/s;

    The satellite orbit was sun -synchronous with apogee 830 km, perigee 640 km, inclination $98.4^{o}$ and orbital period 100 min.

The main operational mode of scientific payload was monitoring mode, when all RELEC instruments were switched on and operated simultaneously. In this mode, data were transmitted to the ground with a rate up to 1.2 Gbyte per day. The total volume of transmitted scientific data was more than 50 Gbyte.

    The RELEC scientific payload included five instruments, i.e. Detector of X, Gamma

Rays and Electrons (DRGE, Russ. acronym), Detector of Ultraviolet (DUV), Telescope-T, Low-frequency Analyzer (NChA, Russ. acronym) – Radio-frequency Analyzer (RChA, Russ. acronym) or PSA – RFA complex and Block Electronics (BE). Their functions are described below

    The DRGE instrument measured X and gamma ray fluxes with intensity up to 20 000

photon/s in the range of detected quantum energy from 0.01 to 3.0 MeV and electron fluxes with intensity up to $2^{16}$ part./s in the energy range from 0.3 to 10.0 MeV.

    The DUV instrument measured ultraviolet A (UV) and red emission in the bands 240 – 400 and 610 – 800 nm, respectively.

    The Telescope-T instrument was used for imaging in the UV and optical ranges.

The NChA – RChA (PSA – RFA) complex of instruments measured electromagnetic-waves in the ULF-ELF-VLF range

    The BE instrument controlled the other scientific instruments, collected and transmitted scientific data. It consisted of a power controller unit and two identical units of scientific data storage (?) a main and reserve ones.

During the space experiment with the RELEC scientific payload the instruments DRGE, DUV, Telescope-T, PSA – RFA were connected to the BE instrument. During operation on the orbit they obtained from BE supply voltage and transmitted to it the collected scientific data. Control of the modes of operation of scientific instruments was also carried out through the BE instrument.

Thus, the RELEC scientific payload provided monitoring measurements of high-energy electron fluxes with a high temporal resolution (better than 15 μs) and the ability to determine the anisotropy of fluxes, as well as the detection of transient atmospheric phenomena in a wide range of the electromagnetic spectrum, from radio to gamma with a temporal resolution better than 15 μs, as well as measurements of electric and magnetic fields in the near-Earth space in the

frequency range from near DC to 15 MHz.

*2.2. The NChA – RChA (PSA – RFA) complex of instruments.*

    The NChA – RChA (PSA – RFA) complex of instruments measured the electromagnetic emission and current in plasma in wide frequency range necessary for the complex study of the

50 processes in ionosphere. These instruments allowed high accuracy measurements of values and fine structure of field variations.





The NChA – RChA (PSA, i.e.SAS3-R – RFA) complex of instruments consisted from the low frequency analyzer NChA (PSA – SAS3-R) and the radio frequency analyzer RChA (RFA).

The NChA (PSA – SAS3-R) instrument consisted of the following units:

- data processing unit for spectral analysis PSA;
- flux gate magnetometer, detector unit (DFM) and electronic unit (BE-FM);
- induction magnetometer IM;
- two identical  complex wave probe (CWZ-1, CWZ-2).

The RChA (RFA) instrument consited of an analyzer unit RFA-E and three-component electric antenna RFA-AE.

The NChA – RChA units were mounted on the outer 3 meters boom (IM, DFM, CWZ-1, CWZ-2, RFA-AE) and on the spacecraft thermostatic panel (units BE-FM, PSA, RFA-E). Their sizes, masses and power consumption are presented in Panasyuk et al., 2016a.

The sketch of mutual displacement of NChA at the boom is presented in Fig. 1. This instrument measured the constant magnetic field by a three-component flux gate magnetometer. The range of measured intensity was no less than ± 64000 nT, non-orthogonality of the meter component was less than 1º, the sampling frequency was 250 Hz. The mutual orthogonality of three measuring axes was provided by the DFM construction

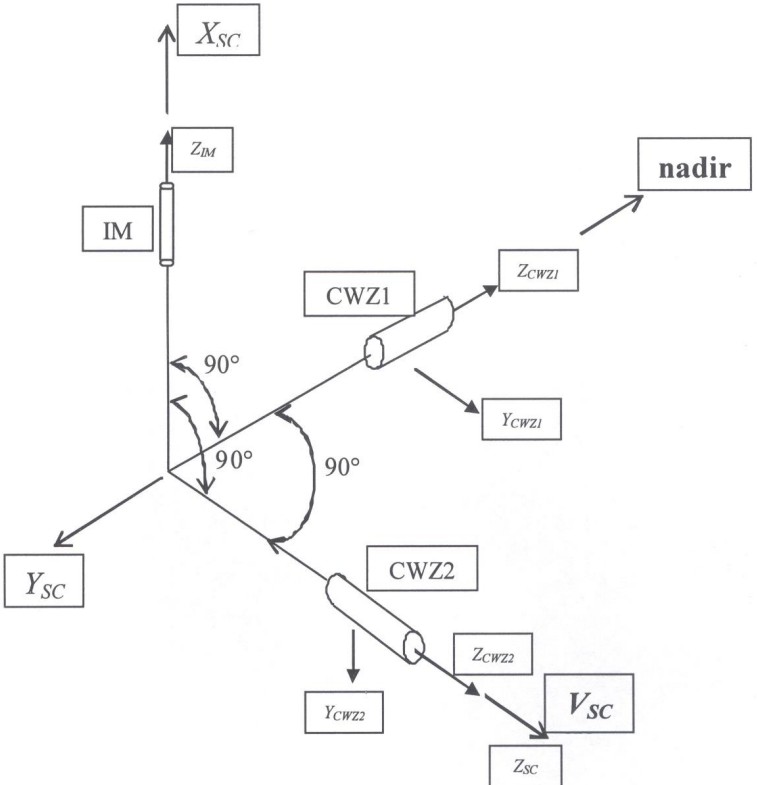

Fig 1. Mutual orientation of PSA – SAS3-R sensors. Xsc, Ysc, Zsc – axis of the spacecraft; Vsc - velocity vector of the spacecraft; Ycwz1, Zcwz1 -  measuring axis of the CWZ1; Ycwz2, Zcwz2 - measuring axis of the CWZ2.



Measurements of values and sign of three components of variable magnetic field induction vector were made with the use of the induction magnetometer IM and magnetic channels of the complex wave probes CWZ-1, CWZ-2. Each of these instruments contained one-component meter of variable magnetic field. To construct an orthogonal coordinate system mutual orthogonality of Z axes of these instruments was provided. The frequency range of the meter was from 0.1 to 40000 Hz.

Measurements of plasma current density were made with the use of current measurement channels of the CVZ-1, CVZ-2 probes. To obtain correct results of measurements these probes were mounted in such a way, that their YOZ planes were parallel to the spacecraft velocity vector.

Measurements of the potential difference were made with the use of electric channels of the CWZ-1, CWZ-2 probes, each of which contained the meter of potential relative the common wire.

The difference of analogue signals from CWZ-1, CWZ-2 was determined in the PSA unit. The signal from CWZ-1 relative to the common wire was also measured there. The common wire was connected with spacecraft via telemetry unit, i.e. via high resistor. Thus the potential difference between CWZ—1 place and spacecraft was determined.

The PSA unit provided:

- producing from the on-board power network $\pm 27$ V of voltage necessary to its own operation as well as of the FM, IM, CWZ-1, CWZ-2 operation, that provided its galvanic isolation from secondary circuits;
- transmission in the digital format of FM, IM, CWZ-1, CWZ-2 outputs;
- calculation of spectral density of measurements values;
- detection of events, i.e. unusual rare electromagnetic phenomena;
- storing measured data before and after event;
- storage of measurement results;
- transmission of measurement results and telemetry data to the ground via spacecraft board systems.

The RFA instrument measured of high frequency emission. It consisted of a analyzer unit, RFA-E and antenna, RFA-AE, which was used to measure the three electric components of electromagnetic field. Physical and technical parameters of RFA instrument units are presented in Panasyuk et al., 2016a.

The RFA instrument measured the three components of electric field, digitized and analyzed the signals in the band from 50 kHz to 15 MHz. Frequency resolution was 10 kHz, temporal resolution was 25 ns. The compressed data were transferred to the spacecraft telemetry system.

The wave form digitization unit contained three 12-bit ADC channels. Each analogue channels included a balancing amplifier with signal shift voltage for balancing the levels with ADC inputs. The digitized signals were stored in an internal buffer. Then the data were processed and compressed by the digital signal processor. Depending on the operational mode and accepted algorithm, there were different types of output signals, such as, wave forms, compressed wave forms, separate wave numbers or the complete spectrum, compressed number of spectra (spectral modes). Calculations were made on FPGA according to the programmed operational modes. The internal memory used either a cyclic buffer or a single-pass FIFO, depending on the operational mode.

### 3. Observation of electromagnetic wave activity on 10.12.2014.

Interesting event in magnetic wave background occurred on December, 10 at 8:54:57 UT, recorded in the magnetic $B_x$ channel. It can be well traced at dynamic spectrograms $\boldsymbol{B_x}$, which are presented in Fig. 2. Particular attention should be paid to the emission with increasing frequency, which was centered at ~ 5 kHz. It is presented in the right panel of Fig. 2.



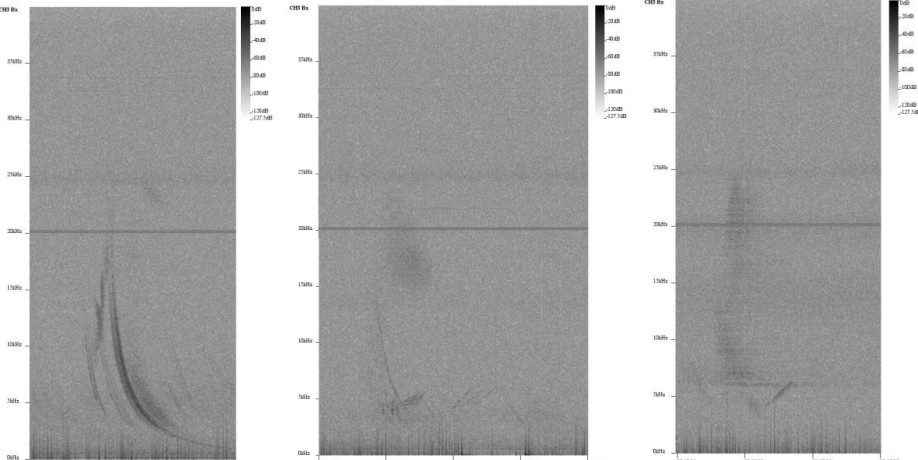

Fig. 2. The NChA instrument. Spectrograms of **Bx** on December 10, 2014.

The NChA instrument operated on 10 December 2014 between 08:54:50.0 - 08:57:10.3 UTC (140 s) in wave form mode in the frequency band 0.1 – 39062.5 Hz of CH0 (**E** electric field intensity) and CH3 (**Bx**-component of magnetic field intensity) channels.

The Vernov satellite orbit parameters 10 December 2014 were: altitude ~ 670 km, local
10   time ~ 21:10 LT, magnetic latitude ~ 75$^0$.

The detailed dynamic spectrogram of **E** signal is presented in Fig. 3. In the frequency range about 10 – 20 kHz the three-wave decay from ~ 20 to ~ 18, ~ 15 and 11 kHz, can be revealed. Two structures at 08:56:20 UT is also can be seen.

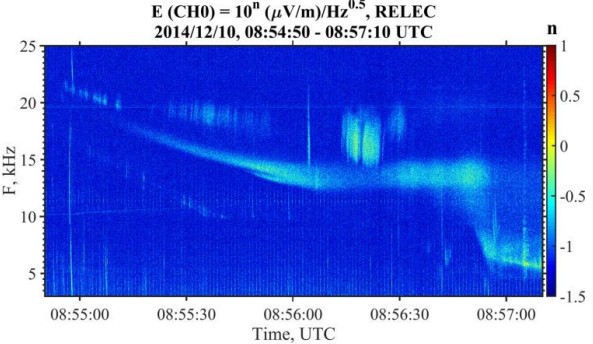

Fig. 3. Dynamic spectrogram of an event.

The Fig. 4 presents spectrogram stretched in time. It confirms the presence of two
20   structures with size ~30 km, where in **E** as band 15 – 17 kHz as narrow-band ~13 kHz emissions are observed. The alternative point may be detection of event with a few second duration and the wave front passed the satellite location.



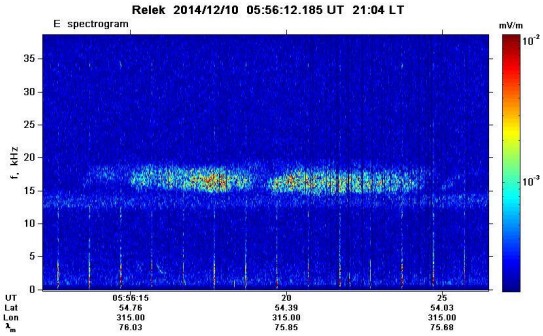

Fig. 4. The same spectrogram as in Fig.3, stretched in time.

5   The three band-pass frequencies, which can reflect two- or may be three-wave decay from ~ 20 to ~ 18, ~ 15 and 11 kHz are also present in the average spectrum of *E* signal, which is shown in Fig. 5.

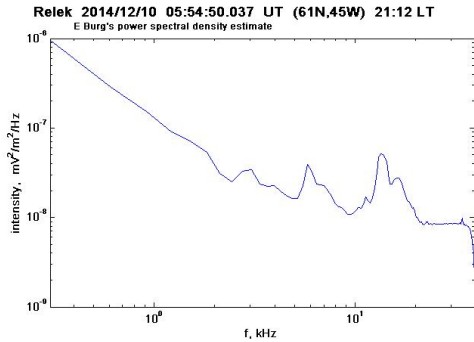

10   Fig. 5 The *E* field spectrum averaged for 08:54:50.0 - 08:57:10.3 UTC interval.

   The wave forms of *E* and *Bx* signals are presented in Fig.6. The wave form of *E* field exhibits EMC noise with 1 Hz frequency, which due to the wide dynamic range on signal amplitude (~120 dB) does not lead to the off-scaling of signal. In the *Bx* signal such noise was

15 not observed clearly, probably as a result of the sensitivity differences of the sensors.

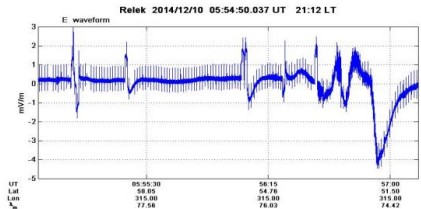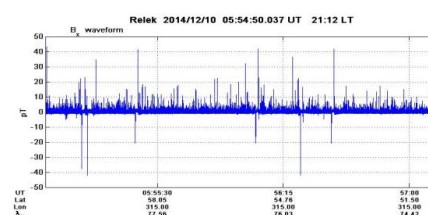

Fig. 6. The wave forms of *E* and *Bx* signals, left panel corresponds to the – *E* field, the right panel corresponds to the *Bx* field. The time scales are UTC - 3 h.

20

   Spectrogram of *E* signal exhibits different types of wave activity. If to use the data about satellite geomagnetic coordinates it is possible to suggest that night sector of main ionospheric dip and longitudinal current region are revealed.





Transformation of **E** frequencies are observed at the end of span ~ 08:56:50 – 08:57:05 UT, where the emission transferring from frequency ~ 13 kHz to ~ 6 kHz taken place (see Fig. 7).

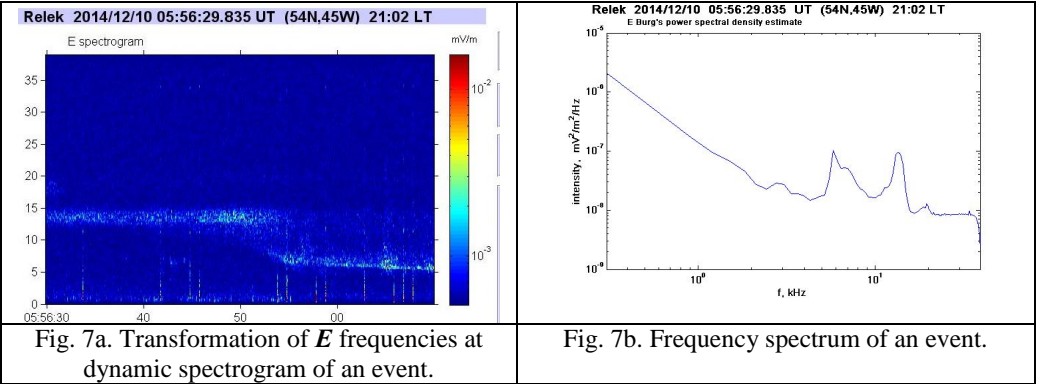

| Fig. 7a. Transformation of **E** frequencies at dynamic spectrogram of an event. | Fig. 7b. Frequency spectrum of an event. |

The component **Bx** undergoes significantly more weak variations than **E** on this orbit. The narrow-band emission on the frequency ~19 kHz, which is possibly generated by navigation transmitters, is observed constantly (see Fig. 2) as in the dynamic as in the averaged spectra.

Discussed above "large-scale" analysis, indicating on the detection by the NChA
10 instrument of geophysical processes, allows consideration of "small-scale" processes.

Let us discuss the increasing on frequency emission, which is centered at ~5 kHz. It is presented in the right panel of the Fig. 2.

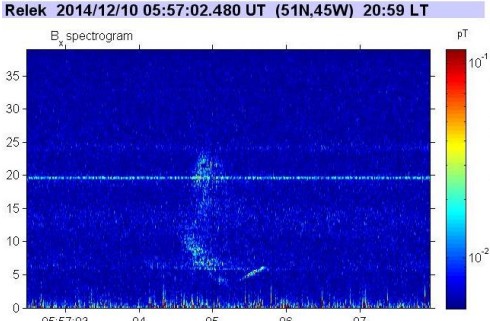

Fig. 8. Detailed spectrogram of of **E.**

In detailed this processes is presented in the Fig. 8, where as in the Fig. 2, after a "noise cloud" 6 – 23 kHz with ~1 s scale the signal increasing on frequency 6 – 7 kHz with ~0.4 s scale
20 is observed. On frequency parameters this signal is similar to the Auroral emission of lasting long enough chorus type. Possibly, due to the fast span of emission area by satellite, only one element of chorus emission was detected. However, strictly linear increase of emission frequency indicates on the man-made nature of this signal without involving data of other RELEC instruments.
25 It is necessary to note, that events detected on the analyzed orbit and presented in the Fig.2, also reflect well such known geophysical emissions as whistlers. The detailed spectrograms of such whistlers are presented in Fig. 9.





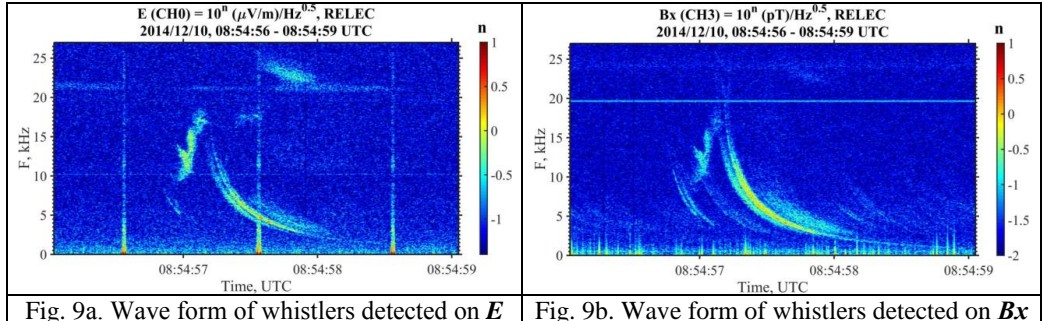

| Fig. 9a. Wave form of whistlers detected on $E$ | Fig. 9b. Wave form of whistlers detected on $Bx$ |
|---|---|

It is noteworthy that in the Fig. 9 a specific rare emission is seen at 08:54:57 UT, which previously was also detected on the Chibis-M microsatellite and named as a "swallow tail", i.e. STW (Ferencz et al., 2010; Klimov et al., 2014).

**4. Discussion.**

The most interesting is the event detected on 10 December 2014 at 8:54:57 UT. The spectrogram of phenomenon presented in the Fig. 8 includes two elements. The first is the signal of increasing frequency in the range 9 – 18 kHz, which similar to the separate chorus element. The second is whistler itself. Duration of the first element is 0.2 s, rate of frequency changing $df/dt = 25$ kHz/s. The signal intensity according to the calibration scale presented at the right side of the figure is about $10^{-\frac{1}{2}}$ µV/m Hz$^{1/2}$ at 10 kHz frequency. As it was noted by Helliwell, 1965, such signals are observed multi studinously at the wave intensity $B \approx 3 \cdot 10^{-3}$ nT/Hz$^{1/2}$.

The origin of such signals according to Trachtengerts and Rycroft, 2011 corresponds to the exiting of trigger signals when energetic electrons occur in the cold plasma region (separate plasma cloud) during the process of its longitudinal drift. After that the cyclotron instability mode is switched-on. This signal can be considered presumably as a precursor of a consequent whistler.

The whistler was analyzed assuming one hop and longitudinal propagation with the method described in Lichtenberger, 2009, Ferencz et al, 2001. The following parameters were obtained: $L = 2.57$, $N_{eq} = 2064$/ccm, $D_0 = 66.4$, $f_n = 18544$ kHz, $f_{heq} = 51234$ kHz. It was assumed $f_{oF2} = 6$ MHz. Here L-shell is the McIllwine parameter of geomagnetic field line; $N_{eq}$ is the electron concentration at the equatorial area of L-shell; $f_{heq}$ is the hyro-frequency of electrons at the equatorial area of L-shell; $f_{oF2}$ is the plasma frequency of F2 layer of ionosphere; $D_0$ is the whistler dispersion; $f_n$ is the nose frequency of whistler.

The causative spheric time is 6.36 s from the beginning of the data. The residuals were average (ms), thus it confirms the inversion. It has to be noted that in the case of a low altitude recordings, like SAS3 data on the Vernov, it is almost impossible to determine the real propagation path of the recorded whistler as if it propagated in a duct, it might well left it above the satellite.

A high-frequency tail at the event (see Fig. 2) reaching 25 kHz indicates the presence of a low conductivity area at the lower ionosphere in the zone of lightning discharges, which contributes to the attenuation of the VLF wave absorption. So, Mikhailov et al., 1997 shown with the use of statistical processing that the growth of high-frequency atmospheric penetration is realized in the region of earthquake before it. The expansion of the atmospheric spectrum to higher frequencies, according to the theory of the properties of the ELF-VLF electromagnetic wave transmission coefficients in the outer ionosphere, indicates a decrease in the conductivity of the lower ionosphere.

In the seismically active period in the preparatory phase of earthquakes a shift in the upper frequency cutoff of the ELF component to higher frequencies and the appearance of the VLF component is observed in the spectra of whistling atmospherics, i.e. attenuation of the ELF-





and VLF-wave decay at their passing into the outer ionosphere from the Earth-ionosphere waveguide takes place.

On the other hand, the presence of thunderstorm activity for a few days or hours before the earthquakes is well known from publications, for example, for the Mediterranean Sea (Ruzhin et al., 2000; Ruzhin et al., 2007) and for the Kamchatka region (Druzhin, 2002). Thus, in the last paper, variations in the noise of natural electromagnetic VLF radiation for the period from January 1997 to December 2001 are considered. It is shown that in Kamchatka a few days before a strong earthquake, high-power pulsed VLF emissions appear in the noise component, which usually cease in a few hours or units of day before the main shock. The experience of the forecast showed that the anomalous phenomena arising in the noise component of the VLF signal can be used for the purposes of a short-term forecast of strong earthquakes.

Sorokin et al., 2011, Sorokin and Ruzhin, 2015 justified theoretically thunderstorm activity above the earthquake preparation area, the possibility of thunderstorm activity activation on the eve of earthquakes was shown, and a corresponding model of its accompanying processes, i.e. EM wave emission up to the VHF band was elaborated.

A map of regional seismic activity after the discussed event is shown in Fig. 10. Each earthquake shown on the map, in principle, could become the source of the observed phenomenon on the Vernov satellite. These seismic regions could be both sources of modification of the ionosphere (that is, contributed to penetration), and the detected whistlers themselves with characteristics and parameters in accordance with Fig. 2.

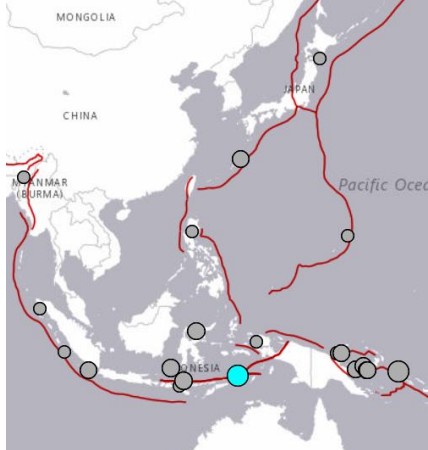

Fig. 10. Earthquakes (its magnitudes M=4.8-6.2) in Southern semi-sphere during 5 days after detection of event presented in Fig. 2.

The possibility of thunderstorm activation before earthquakes was shown by Ruzhin et al., 2000, Ruzhin, Nomicos, 2007. The corresponding model of accompanying processes (Sorokin, Ruzhin, 2015) takes into account emission of electromagnetic waves up to ultra-short wave (USW) range (Sorokin et al., 2011), scattering of USWs (Sorokin et al., 2012) and even over-horizon detection of satellite signals of GPS type (Devi et al., 2012) due to the super-refraction by modified atmosphere at wave propagation under the seismic active area. The main energy of thunderstorm discharges as it is well-known is concentrated in the whistler (spheric) spectral range.

Widening of spectrum of detected spherics in the range of higher frequencies (see Fig. 11) indicates on the lower conductivity of lower ionosphere according to the theory of properties of coefficients of ultra-low frequency (ULF) – very low frequency (VLF) electromagnetic wave passing into the outer ionosphere (Mikhailov et al., 1997). Obtained result allows clear

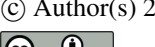


separation of seismic and geomagnetic effects in the ionosphere D-area, which is responsible for ULF – VLF electromagnetic wave passing into the outer ionosphere (see Fig. 2 and in more details Fig. 9). From the last figure it can be seen that high-frequency part of the signal, which was detected on-board of Vernov satellite is higher than 22 kHz and it is strong in these frequencies.

Histograms of spheric maximum frequency distribution are presented in the Fig. 11 for quiet conditions at Kp < 3 (dotted line), but in the seismic active time (thin line), at Kp > 3 in the absence of earthquakes (thick line).

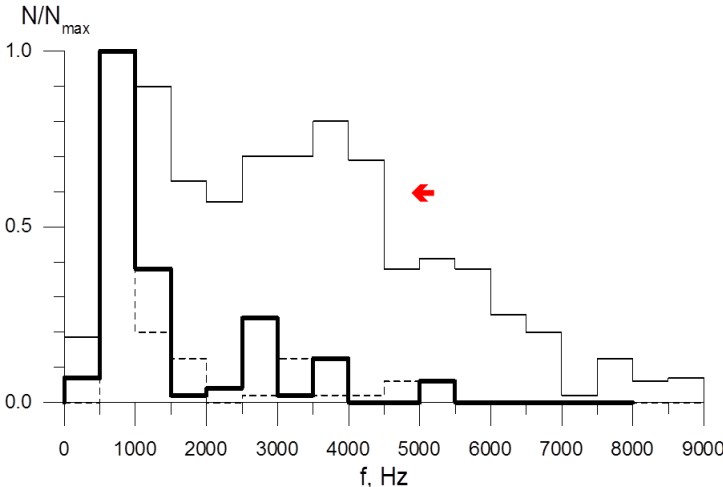

Fig. 11. The histogram of partially dispersed whistler spectra observed on the satellite Intercosmos 24. The dashed line corresponds to the geomagnetic and seismic quiet time, the thick line corresponds to the geomagnetic disturbances, the thin line corresponds to the seismic activity in quiet geomagnetic conditions. The arrow marks the increasing of number of sonograms with frequencies higher than ~5 kHz.

The phenomenon discovered in December 10, 2014 in 08:56:50 UT (see Fig. 4) is also of great interest. It can be suggested that this signal is a part of long whistler (two multiple hops) came from the South semi-sphere. Such whistler long tails (whistler echo) ~60 s was notes by Helliwell, 1965, in which tails with duration up to 30 s were presented. In this case tail had duration 90 s. Possibly, the signal lag had taken place here, such signal lags were discovered by Helliwell when sending Morse code by VLF – transmitter. At the same time, there is reason to suppose that these emissions are the elements of non-linear phenomena excited by packet of low-frequency waves in plasma. Such emissions were observed in Helliwell paper, mentioned above (transmitter in Antarctica, receiver in Canada).

## 5. Conclusion

Non-linear effects in wave activity of two and three wave decay types as well as splitting on to two wave structures were discovered in December 2014 during observations on-board the Vernov satellite. Spherics with specific rare time structure of "swallow tail" type was seen on the obtained spectral diagrams (sonogramsIt were also observed the signals of whistler with long tail type.



## 6. Acknowledgments.

Financial support for this work was provided by the Ministry of Education and Science of Russian Federation, Project № RFMEFI60717X0175. Authors also would like to thank Prof. Janos Lichtenberger for valuable remarks and fruitful discussions.



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
