# Peer review of "NON-LINEAR EFFECTS IN ELECTROMAGNETIC WAVE ACTIVITY OBSERVED IN THE RELEC EXPERIMENT ON-BOARD VERNOV MISSION"

_Annales Geophysicae, 2018_

## Referee Comment (RC1) · Anonymous Referee #1 · 15 Dec 2018

"Non-linear Effects in Electromagnetic Wave Activity Observed in the RELEC Experiment on-board Vernov Mission" by Mikhail I. Panasyuk et al.

Panasyuk et al. showed abnormal amplitude decay of whistler mode waves observed by Vernov satellite. The authors suggested that the abnormal wave decay can be caused by seismic activity. Also, the authors showed a specific wave structure of "swallow tail" type. If these unusual wave phenomena are related to seismic activity as suggested by the authors, I would encourage to validate the relation between these wave phenomena and seismic activity because the authors did not show a clear validation for the relationship. The author must show a clear no relation between the

abnormal wave phenomena and non-seismic activity. I suggest further validation on the wave phenomena in periods for both seismic and non-seismic activities. I think that Section 4 "discussion" is not well discussed and Section 5 "conclusion" is not meaningful (just repeat). I suggest that both sections must be improved with more detailed explanations, discussions, and validation analyses. In my opinion, it requires major revision.

[Specific comments]

Abstract If "RELEC" is an abbreviation/acronym, please define accordingly. The figurative expression "swallowtail type" is not clear in abstract. Please modify using more suitable expression without no figure information.

Discussion As pointed out above, the authors must add the validity analysis for the abnormal wave phenomena related to the seismic activity using a long period including before and after the focusing seismic activity. If the authors do not show the validity analysis, the relation between the wave phenomena and seismic activity is not supported by any demonstration.

Fig. 10 Please describe Fig. 10 in more details. What are the red line, gray and aqua circles?

Fig. 11 Why is Fig. 11 important in this study? The results would be analyzed in a different satellite, different periods, different locations in contrast to the focusing event. If Fig. 11 is important in this study, please show the detailed information on the analysis (data period, observed location, data specifications etc.).

p.11 line 29 What is the nonlinear effect in this study? I could not find discussion on the specific nonlinear effects on the wave phenomena. If a nonlinear effect is important, the author must discuss the specific effects on the nonlinear effects on the wave phenomena.

[Technical corrections and typos] The reviewer mentions that the references on this

study are inadequate. Please add adequate references. (For examples) p.1 line 36 Add a reference on TLEs. p.1 line 41 Add a reference on the effects of high energy particle on electric devices. p.1 line 42 Add a reference on precipitation of high energy particles. p.1 line 54 Add a reference on the wave-particle interactions. p.2 line 3 Add a reference on the whistler mode waves. p.2 line 29 Add a reference on sferics. p.2 line 34 Add a reference on frequency distribution of sferic spectra.

Please add the axis label and the unit for all figures with a sufficient font size. The font size looks like small in the most of present figures. Add the information on satellite location for Figs. 2, 3, 7a, 8, and 9ab as the same with Fig. 4.

p.9, line 13 superscript "1/2"

I hope these comments can help improve this manuscript.

---

## Referee Comment (RC2) · Hayakawa (Referee) · 18 Dec 2018

Referee report on the paper, "Non-linear effects in electromagnetic wave activity observed in the RELEC experiment on board Vernov mission" by Panasyuk et al. submitted to Angeo.

I have read the above paper submitted to ANGEO, and I have found that the mission and its associated experiments seem to be attractive. However, I can recommend rejection of this paper for publication in Angeo. I can list below the reasons why I cannot recommend this paper for publication in Angeo.

(1) English The English of this paper is too bad to understand, sometimes very difficult to understand. It seems that someone (probably the first author) has written this paper, but many other co-authors have never read and revised the text. Extensive revisions of the manuscript among the authors have to be made before submitting a paper to any international journal like Angeo. You need the help of a native English speaker.

(2) Construction (This is the major drawback) This paper is composed of two parts: One is the technical part describing the satellite payload, particle monitoring systems, and wave experiments, and the other, an example of scientific observation results. Both parts are not satisfactory for me and the possible readers. Introduction seems to emphasize the importance of this satellite mission, with the main topic being the precipitation of magnetospheric particles, but there have not been presented any results of particle precipitation in the text. If you want this paper as a technical paper, you have to provide us with the detailed description of the equipment of the satellite mission. Then, when you want to publish a purely scientific paper, you have to delete the technical part (just refer to your previous technical paper) and you have to concentrate yourselves on the results of particle precipitation and the associated wave effects. Instead if you want to give us a new VLF phenomenon, you are highly required to provide us the much more detailed and extensive discussion on the new wave phenomena. However, we cannot find any convincing evidence on the nonlinear wave activity (more detailed analyses on three-wave process). Further, an association of this new wave with seismic activity is not either convincing.

---

## Author Comment (AC1) · 13 Feb 2019

Authors are much pleasured to referees for comprehensive and very useful remarks. According to these remarks we re-elaborated the paper quite significantly. All re-written parts of article are picked-out by yellow in the supplemented \*.pdf file. Below we listed referee's remarks and our commentaries.

"The author must show a clear no relation between the abnormal wave phenomena and non-seismic activity. I suggest further validation on the wave phenomena in periods for both seismic and non-seismic activities. I think that Section 4 "discussion" is not well discussed and Section 5 "conclusion" is not meaningful (just repeat). I suggest that

[Figure]

both sections must be improved with more detailed explanations, discussions, and validation analyses. In my opinion, it requires major revision."

We added the Section 4 "Discussion" by more detailed analysis of wave phenomena in periods of seismic activity. In particular, it was shown by adding of corresponding references that a high-frequency tail at the whistler event reaching 25 kHz indicates the presence of a low conductivity area at the lower ionosphere in the zone of lightning discharges, which contributes to the attenuation of the VLF wave absorption, and that the growth of high-frequency atmospheric penetration is realized in the region of earthquake before it. This was also illustrated by Fig. 11, which was also re-elaborated. References, which show the presence of thunderstorm activity for a few days or hours before the earthquakes, were also added. Thus, it was concluded that analysed whistler may be of lightning origin occurred on the eve of earthquake. Presented in Fig.12 a map of regional seismic activity after the discussed whistler was also significantly redone.

Specific comments

Abstract.

If "RELEC" is an abbreviation/acronym, please define accordingly.

Done – acronym "RELEC" was omitted from Abstract and defined in Section 1 "Introduction"

The figurative expression "swallowtail type" is not clear in abstract. Please modify using more suitable expression without no figure information.

Done - "swallowtail type" was replaced on "which looked like a high frequency tail rising"

4. Discussion.

As pointed out above, the authors must add the validity analysis for the abnormal wave phenomena related to the seismic activity using a long period including before and

after the focusing seismic activity. If the authors do not show the validity analysis, the relation between the wave phenomena and seismic activity is not supported by any demonstration.

For validation of that abnormal wave phenomena may be connected with seismic activity just Fig. 11 is used. It presents partially dispersed whistler spectra in periods of seismic quiet time and of seismic activity. This figure nudicates that increasing of number of whistler events with frequencies higher than ~5 kHz is clearly higher in periods of seismic activity.

Fig. 10 Please describe Fig. 10 in more details. What are the red line, gray and aqua circles?

Fig. 10 was renumbered in Fig. 12. It was re-elaborated significantly. The new map with Earthquakes near Kamchatka region is presented. All lines and marks was described in the text and Figure capture.

Fig. 11 Why is Fig. 11 important in this study? The results would be analyzed in a different satellite, different periods, different locations in contrast to the focusing event. If Fig. 11 is important in this study, please show the detailed information on the analysis (data period, observed location, data specifications etc.).

Fig. 11 is really important for validation of that abnormal wave phenomena may be connected with seismic activity (see above). We think that reference on Intercosmos 24 experiment, which is given in the text is enough for information about this experiment.

p.11 line 29 What is the nonlinear effect in this study? I could not find discussion on the specific nonlinear effects on the wave phenomena. If a nonlinear effect is important, the author must discuss the specific effects on the nonlinear effects on the wave phenomena.

The nonlinear effect in our study means nonlinear dependence of the frequency from time. The Section 4 "Discussion" is added by special spectral analysis based on socalled bi-spectra. Its results are also discused in the Section 5 "Conclusion".

Technical corrections and typos.

The reviewer mentions that the references on this study are inadequate. Please add adequate references. (For examples) p.1 line 36 Add a reference on TLEs.

The reference on TLE is omitted.

p.1 line 41 Add a reference on the effects of high energy particle on electric devices.

Effects of high energy particles are also omitted.

p.1 line 42 Add a reference on precipitation of high energy particles.

Discussion about precipitation of high energy particles is omitted.

p.1 line 54 Add a reference on the wave-particle interactions.

Done

p.2 line 3 Add a reference on the whistler mode waves.

Done

p.2 line 29 Add a reference on sferics.

Done

p.2 line 34 Add a reference on frequency distribution of sferic spectra.

Done

Please add the axis label and the unit for all figures with a sufficient font size. The font size looks like small in the most of present figures.

Done

Add the information on satellite location for Figs. 2, 3, 7a, 8, and 9ab as the same with

Fig. 4.

We add the separate figure with satellite location for all times for which events were discussed in the paper.

p.9, line 13 superscript "1/2"

Done

Please also note the supplement to this comment:
https://www.ann-geophys-discuss.net/angeo-2018-119/angeo-2018-119-AC1-supplement.pdf

**Supplement:**

**NON-LINEAR EFFECTS IN ELECTROMAGNETIC WAVE ACTIVITY OBSERVED IN THE RELEC EXPERIMENT ON-BOARD VERNOV MISSION**

**M.I. Panasyuk[1,2], S.I. Svertilov[1,2], S.I. Klimov[3], V.A. Grushin[3], D.I. Novikov[3], S.P. Savin[3], Yu.Ya. Ruzhin[4], Yu.M. Mikhailov[4], Cs. Ferencz[6], P. Szegedi[7], V.E. Korepanov[5], V.V. Bogomolov[1,2], G.K. Garipov[1], S.V. Belyaev[5], A.N. Demidov[5]**

*1 –Lomonosov Moscow State University, D.V. Skobeltsyn Institute of Nuclear Physics*
*2 –Lomonosov Moscow State University, Physics Department*
*3 – Space Research Institute, Russian Academy of Sciences, Moscow, Russia*
*4 –Pushkov Institute of the Earth magnetism, Ionosphere and radio-wave propagation, Russian Academy of Sciences, Troitsk, Russia*
*5 – Institute of Space Research, Ukrainian Academy of Sciences and National Space Agency, Lviv, Ukraina*
*6 – Eötvös University, Space Research Group, Budapest, Hungary*
*7- BL Electronics Ltd., Solymár, Hungary*

The experiments on-board Vernov satellite were aimed on the study of high energy (relativistic and sub-relativistic) electron acceleration and losses in the trapped radiation areas as well as high altitude electric discharges in the upper Atmosphere. A separate task was study electromagnetic-wave phenomena in the near Earth space and the upper Atmosphere. During observations on December 10, 2014 interesting phenomena were discovered. They are connected to non-linear effects in wave activity of the type of two or three wave decays as well as splitting into two wave structures. Whistlers with specific unusual temporal structure which looked like a high frequency tail rising were observed on the spectral diagrams (sonograms), which were obtained for this time. It was shown that such signals can be caused by seismic activity. The signals of the type of whistler with long tail were also observed. Such signals were also detected by ground stations.

**1. Introduction.**

The aim of scientific experiments with RELEC (acronym *R*elativistic *ELECtr*ons) instruments on-board Vernov satellite (Panasyuk et al., 2016a) was complex study of processes with high energy electrons in the near Earth space, ionosphere and upper Atmosphere including magnetosphere electron precipitation and transient electromagnetic events in the Earth Atmosphere caused by high-altitude atmospheric discharges. Of-coarse these phenomena are tightly connected with wide frequency electromagnetic (EM) waves. Some natural EM phenomena known as space whether and occur in the solar wind – magnetosphere – ionosphere – Earth Atmosphere system generate electromagnetic waves that can be detected in the ionosphere. Whistlers generated by thunderstorm discharges and detected by satellites are typical example of such emissions. Thus, study of EM environment in the near Earth space was also among the main tasks of the RELEC experiment.

It is well-known, that precipitations in the loss cone occur as a result of a change in the pitch-angle distribution of the particles mainly due to wave-particle interaction (see e.g. Lyons, et al., 1972). Calculation of the corresponding electron loss rates requires information about the spectra of the excited waves. Several wave modes generated and propagated in the magnetosphere can interact with relativistic electrons. Among them are whistlers, whistler-mode waves, EM ion cyclotron and electrostatic ion cyclotron waves. The waves are often generated by particles of lower energies, these waves then precipitate particles of higher energies (parasitic diffusion).

Whistler mode waves and electrostatic ion-cyclotron waves scatter most effectively electrons with energies <1 MeV, but they also can scatter particles with higher energies (see e.g. Gurevich, 2007). Electromagnetic ion-cyclotron waves scatter electrons with energies >1 MeV more effectively. Precipitation in the loss cone is connected usually with gyro-resonance interaction of particles with whistler-mode waves, electromagnetic ion-cyclotron (EMIC) waves

and plasmaspheric hisses. This problem was studied in details, see e.g. Kennel and Petscheck, 1966; Horne and Thorne, 2003; Thorne et al., 1977; Summers and Thorne, 2003; Albert, 2003; Meredith et al., 2006; Shprits et al., 2008 a,b and Ferencz et al., 2001.

Atmospherics or spherics are electromagnetic signals produced by lightning discharges (see e.g. Roussel- Dupré, et al., 2008.). The average occurrence of lightnings is about 100 strikes per second. A lightning discharge consists of two stages. There are pre-discharges and main discharges, which differ by current and spectrum of emitted radio-waves. Ultra-long waves are emitted in the main discharge, while long, middle and even short waves are emitted in the pre-discharge.

The maximum of spheric energy lies in the range of about 4-8 kHz (see e.g. Ferencz, et al. 2010.). If spherics are produced by nearby thunderstorms, their spectrum is determined only by the emission spectrum of thunderstorm discharge. If the source is a distant thunderstorm, its spectrum is determined also by the conditions of radio-wave propagation from the thunderstorm to the radio receiver. Spherics have a weak attenuation and can propagate over long distances.

Spherics may penetrate into the ionosphere and propagate along the magnetic field line, reaching the Earth-Ionosphere waveguide again in the other hemisphere and can be recorded on the ground. These waves exhibit frequency changes versus time and called whistlers. Their peculiarity is associated with Very Low Frequency wave propagation in magneto-ionic medium, see e.g. Hellivell, 1965; Ferencz et al. 2007, 2009; Ferencz et al., 2001.

The type of spheric spectrum is determined by magnetic field intensity and electron concentration along trajectory. The spectrum covers frequencies from hundreds Hz up to 20-30 kHz (Klimov et al., 2014). The spheric property analysis allows us to determine the electron concentration distribution along the propagation path, see e.g. Park, 1972; Lichtenberger, 2009. A rare phenomenon called knee whistlers (Carpenter, 1963) may be used to determine the location of the plasmapause. Low-frequency branches of the spheric spectrum (ion whistlers) were detected on the satellites at frequencies below 400-500 Hz, over which the relative concentrations of ions and electrons, as well as other parameters of the ionosphere can be determined.

ULF-ELF-VLF electromagnetic radiation plays the same role for the study of plasma processes in space as seismic waves for the Earth structure study. In comparison with EM processes in other environments, waves in plasma have a number of definite characteristic peculiarities. The resonance effect is the most important. It occurs due to the wave-particle interaction, wave transformations, resonator and waveguide formation. Due to the resonance effect, ultra-low frequency waves provide information about dynamic phenomena in the near-Earth space and the upper Atmosphere. By this they reach sufficiently large amplitudes to have significant influence on the plasma fluxes and effectively accelerate electrons in the magnetosphere. It was shown recently, that not only magnetosphere – ionosphere phenomena are accompanied by plasma disturbance, but also ground geophysical ones caused by large energy release such as explosions, hurricanes, thunderstorms and earthquakes.

Emission intensity increasing in the ELF-VLF bands observed before earthquakes on many satellites in the narrow-band detection mode. However, to the present time, the nature of this effect remains controversial. Our cycle of work using broadband and narrowband observations on board satellites has allowed us to obtain previously unknown data on this effect. Thus, the first detection of broadband ELF-VLF emissions were performed on the satellite Intercosmos-24 during its fly for three hours over the area of the Iranian earthquake since the beginning of the June, 20 at 21:00:07,1 UT (Mikhailov et al., 1991). The observed nature of the emission was not noise, as indicated in other studies, but discrete with the parameters of typical partially dispersed whistler. It was discovered that their follow-up frequency was abnormally high and intensity was higher in comparison with usually rarely observed signals in morning hours of local time at similar heights and latitudes. Also the big extent on latitude of their detection zone was found.

The other intriguing problem is study of non-linear phenomena in EM environment. Due to the exponentially rapid change in the concentration of neutral molecules with height in the Atmosphere and the presence of the Earth's magnetic field, the properties of a free ionospheric plasma are extremely diverse. A large number of different waves can exist in a plasma in a magnetic field, which causes an extremely large variety of nonlinear phenomena in the ionospheric plasma (Gurevich, et al. 2007). In the case of weak nonlinearity, the main nonlinear wave process in plasma is three-wave resonance, for which the conservation laws describing such processes must be satisfied, i.e. one wave splits into two waves, two waves merge into one wave. The references also one can find in (Savin, et al, 2014).

Interesting EM wave phenomena possible connected with seismic activity and non-linear processes were observed during the RELEC Vernov experiment on December 10, 2014. Detailed analysis of these events will be presented below in subsequent sections of this article.

**2. Magnetic-wave instruments PSA – RFA as a part of the RELEC scientific payload on-board Vernov satellite.**

The RELEC scientific payload described in details by Panasyuk et al., 2016 b,c was operated on-board small spacecraft Vernov between July – December, 2014. It was named in honor of pioneer of space research Soviet academician Sergey Nikolaevich Vernov. The satellite orbit was sun -synchronous with apogee 830 km, perigee 640 km, inclination $98.4^{o}$ and orbital period 100 min.

The EM emission and current in plasma in wide frequency range were measured by NChA – RChA (PSA – RFA) complex. These instruments allowed high accuracy measurements of values and fine structure of field variations.

The NChA – RChA (PSA, i.e.SAS3-R – RFA) complex of instruments consisted from the low frequency analyzer NChA (PSA – SAS3-R) and the radio frequency analyzer RChA (RFA).

The NChA (PSA – SAS3-R) instrument consisted of the following units:
- data processing unit for spectral analysis PSA;
- flux gate magnetometer, detector unit (DFM) and electronic unit (BE-FM);
- induction magnetometer IM;
- two identical complex wave probe (CWZ-1, CWZ-2).

The RChA (RFA) instrument consited of an analyzer unit RFA-E and three-component electric antenna RFA-AE.

The NChA – RChA units were mounted on the outer 3 meters boom (IM, DFM, CWZ-1, CWZ-2, RFA-AE) and on the spacecraft thermostatic panel (units BE-FM, PSA, RFA-E). Their sizes, masses and power consumption are presented in Panasyuk et al., 2016a.

The sketch of mutual displacement of NChA at the boom is presented in Fig. 1. This instrument measured the constant magnetic field by a three-component flux gate magnetometer. The range of measured intensity was no less than $\pm$ 64000 nT, non-orthogonality of the meter component was less than 1º, the sampling frequency was 250 Hz. The mutual orthogonality of three measuring axes was provided by the DFM construction

Measurements of values and sign of three components of variable magnetic field induction vector were made with the use of the induction magnetometer IM and magnetic channels of the complex wave probes CWZ-1, CWZ-2. Each of these instruments contained one-component meter of variable magnetic field. To construct an orthogonal coordinate system mutual orthogonality of Z axes of these instruments was provided. The frequency range of the meter was from 0.1 to 40000 Hz.

Measurements of plasma current density were made by current measurement channels of the CVZ-1, CVZ-2 probes. In order to obtain correct results of measurements these probes were mounted in such a way, that their YOZ planes were parallel to the spacecraft velocity vector.

[Figure]

Fig. 1. Mutual orientation of PSA – SAS3-R sensors. Xsc, Ysc, Zsc – axes of the spacecraft; Vsc - velocity vector of the spacecraft; Ycwz1, Zcwz1 - measuring axes of the CWZ1; Ycwz2, Zcwz2 - measuring axes of the CWZ2.

Measurements of the potential difference were made with the use of electric channels of the CWZ-1, CWZ-2 probes, each contained the meter of potential relative the common wire.

The difference of analogue signals from CWZ-1, CWZ-2 was determined in the PSA unit. The signal from CWZ-1 relative to the common wire was also measured there. The common wire was connected with spacecraft via telemetry unit, i.e. via high resistor. Thus the potential difference between CWZ—1 place and spacecraft was determined.

The PSA unit provided:

- producing from the on-board power network ± 27 V of voltage necessary to its own operation as well as of the FM, IM, CWZ-1, CWZ-2 operation, that provided its galvanic isolation from secondary circuits;

- transmission in the digital format of FM, IM, CWZ-1, CWZ-2 outputs;

- calculation of spectral density of measurements values;

- detection of the events, i.e. unusual rare electromagnetic phenomena;

- storing measured data before and after the event;

- storage of measurement results;

- transmission of measurement results and telemetry data to the ground via spacecraft board systems.

The NChA (PSA – SAS3-R) instrument operated in three main operation modes and a command controlled recording frequency bandwidth. The main operation modes were the event

detection mode, the monitoring mode and the continuous wave recording (burst) form mode (means digitization frequency 80 kHz). The recording frequency bandwidth was from <0.1 Hz to 40 kHz. This is approximately two times wider than the frequency bandwidth of the most part of earlier ionosphere satellite experiments, except the CHIBIS-M micro-satellite electromagnetic wave measurements, which has also similar up to 40 kHz wide band VLF recording possibilities (Zeleniy et al., 2014).

The RFA instrument measured high frequency emission. It consisted of an analyzer unit, RFA-E and antenna, RFA-AE, which was used to measure three electric components of the electromagnetic field. Physical and technical parameters of RFA instrument units are presented in Panasyuk et al., 2016a.

The RFA instrument measured three components of electric field, digitized and analyzed the signals in the band from 50 kHz to 15 MHz. Frequency resolution was 10 kHz, temporal resolution was 25 ns. The compressed data were transferred to the spacecraft telemetry system.

The wave form digitization unit contained three 12-bit ADC channels. Each analogue channels included a balancing amplifier with signal shift voltage for balancing the levels with ADC inputs. The digitized signals were stored in an internal buffer. Then the data were processed and compressed by the digital signal processor. Depending on the operational mode and accepted algorithm, there were different types of output signals, such as, wave forms, compressed wave forms, separate wave numbers or the complete spectrum, compressed number of spectra (spectral modes). Calculations were made on FPGA according to the programmed operational modes. The internal memory used either a cyclic buffer or a single-pass FIFO, depending on the operational mode.

**3. Observation of electromagnetic wave activity on 10.12.2014.**

Interesting events in magnetic wave background occurred on December, 10 at 8:54:57 UT, were recorded in the magnetic $B_x$ channel of NChA (PSA – SAS3-R) instrument, which operated on December 2014, 10 between 08:54:50.0 - 08:57:10.3 UTC (140 s) in wave recording form mode in the frequency band 0.1 – 39062.5 Hz of CH0 ($E$ electric field intensity) and CH3 ($Bx$-component of magnetic field intensity) channels.

The footprint of the Vernov satellite orbit at the above mentioned time interval is presented in Fig. 2.

[Figure]

Fig. 2. The Vernov satellite orbit, 10 December 2014, altitude ~ 670 km, local time ~ 21:10 LT, magnetic latitude ~ $75^0$.

A sequence of unusual events was revealed on the spectral diagrams (sonograms), which were obtained for this time. It can be well traced at $B_x$ dynamic spectrograms obtained, which

are presented in Fig. 3. The whistler with specific temporal structure can be seen in the left panel. This whistler accompanied by non-linear process in wave activity, when initial EM wave with given frequency underwent decay on two or three wave modes and then two wave structures occurred. Particular attention should be paid to the emission with increasing frequency, which was centered at ~ 5 kHz. It is presented in the right panel of Fig. 3.

[Figure]

Fig. 3. The NChA instrument. Spectrograms of $B_x$ on December 10, 2014.

The detailed time structure of the whistler can be analyzed from spectrograms obtained in wave recording mode for electric field ($E$) and one magnetic field component ($B_x$) are presented in Fig. 3. It is noteworthy that a specific rare emission is seen at 08:54:57 UT, which looked like a high frequency tail rising from about 10 to 15 kHz. Previously similar structures were also detected on the Chibis-M microsatellite and named as a "swallow tail" whistler, i.e. STW (Ferencz et al., 2010; Klimov et al., 2014).

[Figure]

| Fig. 4a. Wave form of whistlers detected on $E$. | Fig. 4b. Wave form of whistlers detected on $B_x$. |

The dynamic spectrogram of $E$ signal for a longer time interval 08:54:50 – 08:57:10 UT is presented in Fig. 5. There are could be seen three sloping lines as if coming out of one point in upper left corner of the panel at about 20 kHz. Then these lines diverge by to ~18, ~15 and ~11 kHz. The possible interpretation of such spectral evolution may be three-wave decay from ~ 20 to ~ 18, ~ 15 and 11 kHz. After that two clear structures at 08:56:20 UT are formed.

[Figure]

Fig. 5. Dynamic spectrogram of the event $E$ signal.

The spectrogram stretched in time is presented in Fig. 6. It confirms the presence of two structures with size ~30 km, whereas band 15 – 17 kHz as narrow-band ~13 kHz emissions in $E$ are observed.

[Figure]

Fig. 6. The same spectrogram as in Fig.5 ($E$ signal), stretched in time.

[revised manuscript text omitted]

Thus, whistler may be of lightning origin occurred on the eve of the earthquake. Above the epicenter of the earthquake, a "window" is formed that ensures the penetration of high-frequency radiation through the ionosphere.

A map of regional seismic activity after the discussed event is shown in Fig. 12. Each earthquake shown on the map, in principle, could become the source of the observed phenomenon on the Vernov satellite. These seismic regions could be both sources of modification of the ionosphere (that is, contributed to penetration), and the detected whistlers themselves with characteristics and parameters in accordance with Fig. 2.

During 5 days (December 11 – 15, 2014) after satellite flying near abnormal whistler detection region 9 earthquakes with M≥4.0 occurred (see in map area). By this, 5 of them took place during two days after Vernov satellite flying (December 11 - 15, 2014). All earthquakes are marked by circles.

Earthquake parameters occurring within two days later:
1.      Three earthquakes (area 1, 51°N/179°E) with magnitudes M = 4.1- 4.7 occurred a day later, that is, 11.12.2014 from 9:26 to 20:30 UT.
2.      Two earthquakes (area 2, 54.3°N/166.2°E) with magnitudes greater than M = 4.5, two days later (12.12.2014, 18:17 - 21:30 UT).

[Figure]

Fig. 12. Earthquakes (magnitudes M≥4.0) during 5 days after detection of the event presented in Fig. 2. Brown lines are active faults of the Earth's crust or localization of possible earthquakes, as well as sources of their precursors (lightning discharges and abnormal whistlers). A cross marked the position of the Vernov satellite at the time of registration of the phenomenon under discussion-whistler with a frequency exceeding 25 kHz.

All parts of the fault in the vicinity of the whistler detection point including areas of seismic activity 1 and 2 (see Fig. 12) are at the stage of earthquake preparation, that is, the accumulated energy can be realized in the form of earthquakes anywhere in the fault. As a result, the "window" of transparency in the ionosphere may be looked like a strip stretched along the fault. The its bandwidth at the ionosphere level is determined by the radius of an isolated earthquake preparation from the Dobrovolsky formula, adapted for ionospheric precursors (see Ruzhin and Depueva, 1996; Oraevsky et al., 1995 for details) as $R$[km] = exp$M$, where $M$ is the magnitude of the coming earthquake. For magnitudes of earthquakes that occurred within 2 days after the whistler detection on the satellite orbit, shown in Fig.12 (magnitudes M4.3-M4.7, see areas 1-2), the transparency stripe bandwidth will be $2R$ or 150-220 km.

Atmospheric discharges in the preparation zone (Ruzhin at al., 2000; Ruzhin at al., 2007; Sorokin et al., 2011; Sorokin et al., 2012) on the eve of an earthquake (potential sources of whistlers emission) occur spontaneously in time and mosaic in space, and in our case within the band determined by the fault configuration. This is valid for both earthquakes and seaquakes (Ruzhin at al., 2000) with epicenters under the bottom of the sea. Thus, the presence of the strip band of transparency and lightning discharges in the fault zone is a highly likely scenario for the appearance of an anomalous whistle detected on December 10, 2014 on the orbit of the Vernov satellite (Fig. 3).

These seismic regions could be sources of both modifications of the ionosphere (which contributed to the penetration of high-frequency whistles) in the area along the fault (red line in Fig.12), and the whistlers detected on the satellite.

The phenomenon discovered on December 10, 2014 in 08:56:50 UT (see Fig. 5) is also of great interest. It can be suggested that this signal is a part of long whistler (two multiple hops) came from the South semi-sphere. Such whistler long tails (whistler echo) ~60 s was noted by Helliwell, 1965, in which tails with duration up to 30 s were presented. In this case tail had duration 90 s. Possibly, the signal lag had taken place here, such signal lags were discovered by Helliwell when sending Morse code by VLF – transmitter. At the same time, there is reason to suppose that these emissions are the elements of non-linear phenomena excited by packet of low-frequency waves in plasma. Such emissions were observed in Helliwell paper, mentioned above (transmitter in Antarctica, receiver in Canada).

Nonlinear dependence of the frequency from time is defined by the non-uniform magnetosphere parameters. To validate the possible three-wave decay, indication on which could be seen from Fig. 5, we processed corresponding wave form by so-called bi-spectrum technique (see Savin, et al, 2014 and references therein). As the result, we didn't find any three-wave interactions at ~ 08:55:30, 08:56:20 and 08:56:50 UT, while 2-3 changing signals are evident in Figs 5, 6 and 8. It means that the signals were detected far from their sources, where a generation occurs. However, we found a spike at the beginning of event presented in Figs 5 and 8. This spike is presented in Fig. 13. Form this figure one can see several spectral maxima inside the spike, one of which with smaller amplitude can be revealed over the all-time interval, i.e. from 08:54:00 – 08:54:21, 424 UTC. For this spike we made bi- spectral analysis, the corresponding bi spectrum is presented in Fig. 14.

[Figure]

Fig. 13. Spectrogram of the burst at the beginning of event presented in Figs 5 and 8 (***E*** field).

Fig. 14. Bi-spectrum for the Fig. 13 spike.

Bi- spectra shows the coherence in a three-wave nonlinear processes when $F1 + F2 = F3$ ($F1$ – horizontal axis, $F2$ – vertical one). As it can be seen from Fig. 14, the most prominent maximum (red point) is revealed at ~ 1300 Hz (vertical axis), it corresponds to the generation of the 2nd harmonic at ~ 2700 Hz. Probably, at ~ 3900 Hz the 3rd harmonic is also seen. From the obtained bi-spectrum it could be concluded that for 2nd and 3rd harmonics $F1 + F2 = F3 = $ const. It could be interpreted as decay of non-linear wave on two daughter waves. The presence of persisting vertical frequency at 3rd harmonic indicates on that decay process on summary frequency changes on the process with fixed frequency $F1 = $ const, which could be associated with cascade of non-linear waves (Savin et al., 2014).

**5. Conclusion**

Non-linear effects in wave activity of two and three wave decay types as well as splitting on to two wave structures were discovered in December 2014 during observations on-board the Vernov satellite. Spherics with specific rare time structure of "swallow tail" type was seen on the obtained spectral diagrams (sonograms). The signals of whistler with long tail type were also observed.

The possibility of thunderstorm activation before earthquakes was shown by Ruzhin et al., 2000; Yamada et al., 2002; Ruzhin, Nomicos, 2007. The corresponding model of accompanying processes (Sorokin, Ruzhin, 2015) takes into account emission of electromagnetic waves up to ultra-short wave (VHF) range (Sorokin et al., 2011), scattering of VHF waves (Sorokin et al., 2012) and even over-horizon detection of satellite signals of GPS type (Devi et al., 2012) due to the super-refraction by modified Atmosphere at wave propagation under the seismic active area. It is well-known that the main energy of thunderstorm discharges is concentrated in the whistler (spheric) spectral range.

Widening of spectrum of detected spherics in the range of higher frequencies (see Fig. 10) indicates on the lower conductivity of lower ionosphere according to the theory of properties of coefficients of ultra-low frequency (ULF) – very low frequency (VLF) electromagnetic wave passing into the outer ionosphere (Mikhailov et al., 1997a). Obtained result allows clear separation of seismic and geomagnetic effects in the ionosphere D-area, which is responsible for ULF – VLF electromagnetic wave passing into the outer ionosphere (see Fig. 2 and in more details Fig. 3). From the last figure it can be seen that high-frequency part of the signal, which was detected on-board Vernov satellite is higher than 25 kHz and is strong in these frequencies.

Note that such coincidence of the place and time of the seismically active region and the Vernov satellite (point X - above the fault between the Commander and Aleutian Islands) allows us to interpret the high-frequency whistle (Fig. 2) as a possible precursor of future seismic activity in the fault area or, specifically, listed earthquakes.

Similar to (Savin, et al, 2014), we also conclude that the nonlinear interactions are characteristic also to the ionospheric phenomena. Especially it concerns the harmonic generation and decay processes at the fixed sum frequency. The bi- coherence level of ~ 10-30% is not high enough to prove that the spacecraft is measuring in the physical interaction region. But it looks like that the registered nonlinear waves have still the amplitudes which provide strong nonlinear interactions (3, 4 waves, etc. it should be clarified further). Observation of non-linear effects far from the source in the Atmosphere indicates on very high intensities of non-linear waves on the source, which can be connected with electric discharge. In turn, it indicates on the presence of very strong electromagnetic fields, which provided conditions for non-linear wave processes.

**6. Acknowledgments.**

Financial support for this work was provided by the Ministry of Education and Science of Russian Federation, Project № RFMEFI60717X0175. Authors also would like to thank Prof. Janos Lichtenberger for valuable remarks and fruitful discussions.

---

## Author Comment (AC2) · 13 Feb 2019

Authors are much pleasured to referees for comprehensive and very useful remarks. According to these remarks we re-elaborated the paper quite significantly. All re-written parts of article are picked-out by yellow in the supplemented \*.pdf file. Below we listed referee's remarks and our commentaries.

English The English of this paper is too bad to understand, sometimes very difficult to understand.

We try to improve English

[Figure]

This paper is composed of two parts: One is the technical part describing the satellite payload, particle monitoring systems, and wave experiments, and the other, an example of scientific observation results. Both parts are not satisfactory for me and the possible readers. Introduction seems to emphasize the importance of this satellite mission, with the main topic being the precipitation of magnetospheric particles, but there have not been presented any results of particle precipitation in the text.

We re-wrote Introduction. Discussion about precipitation was omitted otherwise more detailed analysis of wave phenomena was added.

If you want this paper as a technical paper, you have to provide us with the detailed description of the equipment of the satellite mission. Then, when you want to publish a purely scientific paper, you have to delete the technical part (just refer to your previous technical paper) and you have to concentrate yourselves on the results of particle precipitation and the associated wave effects. Instead if you want to give us a new VLF phenomenon, you are highly required to provide us the much more detailed and extensive discussion on the new wave phenomena.

We cut the Section 2. Only necessary description of wave instruments was leaved.

However, we cannot find any convincing evidence on the nonlinear wave activity (more detailed analyses on three-wave process).

We added into the Section 4 "Discussion" analysis of non-linear wave activity based on so-called bi-spectra and discussion of its results.

Further, an association of this new wave with seismic activity is not either convincing.

We added more detailed analysis as of association of observed new wave with seismic activity as of nonlinear wave activity into the Section 4 "Discussion" (see our commentary on the remarks of Referee 1).

Please also note the supplement to this comment:

https://www.ann-geophys-discuss.net/angeo-2018-119/angeo-2018-119-AC2-supplement.pdf

[Figure]

**Supplement:**

**NON-LINEAR EFFECTS IN ELECTROMAGNETIC WAVE ACTIVITY OBSERVED IN THE RELEC EXPERIMENT ON-BOARD VERNOV MISSION**

**M.I. Panasyuk1,2, S.I. Svertilov1,2, S.I. Klimov3, V.A. Grushin3, D.I. Novikov3, S.P. Savin3, Yu.Ya. Ruzhin4, Yu.M. Mikhailov4, Cs. Ferencz6, P. Szegedi7, V.E. Korepanov5, V.V. Bogomolov1,2, G.K. Garipov1, S.V. Belyaev5, A.N. Demidov5**

 1 –Lomonosov Moscow State University, D.V. Skobeltsyn Institute of Nuclear Physics 2 –Lomonosov Moscow State University, Physics Department 3 – Space Research Institute, Russian Academy of Sciences, Moscow, Russia
4 –Pushkov Institute of the Earth magnetism, Ionosphere and radio-wave propagation, Russian Academy of Sciences, Troitsk, Russia

5 – Institute of Space Research, Ukrainian Academy of Sciences and National Space Agency, Lviv, Ukraina 6 – Eötvös University, Space Research Group, Budapest, Hungary

7- BL Electronics Ltd., Solymár, Hungary

The experiments on-board Vernov satellite were aimed on the study of high energy (relativistic and subrelativistic) electron acceleration and losses in the trapped radiation areas as well as high altitude electric discharges in the upper Atmosphere. A separate task was study electromagnetic-wave phenomena in the near Earth space and the upper Atmosphere. During observations on December 10, 2014 interesting phenomena were discovered. They are connected to non-linear effects in wave activity of the type of two or three wave decays as well as splitting into two wave structures. Whistlers with specific unusual temporal structure which looked like a high frequency tail rising were observed on the spectral diagrams (sonograms), which were obtained for this time. It was shown that such signals can be caused by seismic activity. The signals of the type of whistler with long tail were also observed. Such signals were also detected by ground stations.

**1. Introduction.**

The aim of scientific experiments with RELEC (acronym Relativistic *ELECtrons*) instruments on-board Vernov satellite (Panasyuk et al., 2016a) was complex study of processes with high energy electrons in the near Earth space, ionosphere and upper Atmosphere including magnetosphere electron precipitation and transient electromagnetic events in the Earth Atmosphere caused by high-altitude atmospheric discharges. Of-coarse these phenomena are tightly connected with wide frequency electromagnetic (EM) waves. Some natural EM phenomena known as space whether and occur in the solar wind – magnetosphere – ionosphere – Earth Atmosphere system generate electromagnetic waves that can be detected in the ionosphere. Whistlers generated by thunderstorm discharges and detected by satellites are typical example of such emissions. Thus, study of EM environment in the near Earth space was also among the main tasks of the RELEC experiment.

It is well-known, that precipitations in the loss cone occur as a result of a change in the pitch-angle distribution of the particles mainly due to wave-particle interaction (see e.g. Lyons, et al., 1972). Calculation of the corresponding electron loss rates requires information about the spectra of the excited waves. Several wave modes generated and propagated in the magnetosphere can interact with relativistic electrons. Among them are whistlers, whistler-mode waves, EM ion cyclotron and electrostatic ion cyclotron waves. The waves are often generated by particles of lower energies, these waves then precipitate particles of higher energies (parasitic diffusion).

Whistler mode waves and electrostatic ion-cyclotron waves scatter most effectively electrons with energies <1 MeV, but they also can scatter particles with higher energies (see e.g. Gurevich, 2007). Electromagnetic ion-cyclotron waves scatter electrons with energies >1 MeV more effectively. Precipitation in the loss cone is connected usually with gyro-resonance interaction of particles with whistler-mode waves, electromagnetic ion-cyclotron (EMIC) waves

and plasmaspheric hisses. This problem was studied in details, see e.g. Kennel and Petscheck, 1966; Horne and Thorne, 2003; Thorne et al., 1977; Summers and Thorne, 2003; Albert, 2003; Meredith et al., 2006; Shprits et al., 2008 a,b and Ferencz et al., 2001.

Atmospherics or spherics are electromagnetic signals produced by lightning discharges (see e.g. Roussel- Dupré, et al., 2008.). The average occurrence of lightnings is about 100 strikes per second. A lightning discharge consists of two stages. There are pre-discharges and main discharges, which differ by current and spectrum of emitted radio-waves. Ultra-long waves are emitted in the main discharge, while long, middle and even short waves are emitted in the pre-discharge.

The maximum of spheric energy lies in the range of about 4-8 kHz (see e.g. Ferencz, et al. 2010.). If spherics are produced by nearby thunderstorms, their spectrum is determined only by the emission spectrum of thunderstorm discharge. If the source is a distant thunderstorm, its spectrum is determined also by the conditions of radio-wave propagation from the thunderstorm to the radio receiver. Spherics have a weak attenuation and can propagate over long distances.

Spherics may penetrate into the ionosphere and propagate along the magnetic field line, reaching the Earth-Ionosphere waveguide again in the other hemisphere and can be recorded on the ground. These waves exhibit frequency changes versus time and called whistlers. Their peculiarity is associated with Very Low Frequency wave propagation in magneto-ionic medium, see e.g. Hellivell, 1965; Ferencz et al. 2007, 2009; Ferencz et al., 2001.

The type of spheric spectrum is determined by magnetic field intensity and electron concentration along trajectory. The spectrum covers frequencies from hundreds Hz up to 20-30 kHz (Klimov et al., 2014). The spheric property analysis allows us to determine the electron concentration distribution along the propagation path, see e.g. Park, 1972; Lichtenberger, 2009. A rare phenomenon called knee whistlers (Carpenter, 1963) may be used to determine the location of the plasmapause. Low-frequency branches of the spheric spectrum (ion whistlers) were detected on the satellites at frequencies below 400-500 Hz, over which the relative concentrations of ions and electrons, as well as other parameters of the ionosphere can be determined.

ULF-ELF-VLF electromagnetic radiation plays the same role for the study of plasma processes in space as seismic waves for the Earth structure study. In comparison with EM processes in other environments, waves in plasma have a number of definite characteristic peculiarities. The resonance effect is the most important. It occurs due to the wave-particle interaction, wave transformations, resonator and waveguide formation. Due to the resonance effect, ultra-low frequency waves provide information about dynamic phenomena in the near-Earth space and the upper Atmosphere. By this they reach sufficiently large amplitudes to have significant influence on the plasma fluxes and effectively accelerate electrons in the magnetosphere. It was shown recently, that not only magnetosphere – ionosphere phenomena are accompanied by plasma disturbance, but also ground geophysical ones caused by large energy release such as explosions, hurricanes, thunderstorms and earthquakes.

Emission intensity increasing in the ELF-VLF bands observed before earthquakes on many satellites in the narrow-band detection mode. However, to the present time, the nature of this effect remains controversial. Our cycle of work using broadband and narrowband observations on board satellites has allowed us to obtain previously unknown data on this effect. Thus, the first detection of broadband ELF-VLF emissions were performed on the satellite Intercosmos-24 during its fly for three hours over the area of the Iranian earthquake since the beginning of the June, 20 at 21:00:07,1 UT (Mikhailov et al., 1991). The observed nature of the emission was not noise, as indicated in other studies, but discrete with the parameters of typical partially dispersed whistler. It was discovered that their follow-up frequency was abnormally high and intensity was higher in comparison with usually rarely observed signals in morning hours of local time at similar heights and latitudes. Also the big extent on latitude of their detection zone was found. The other intriguing problem is study of non-linear phenomena in EM environment. Due to the exponentially rapid change in the concentration of neutral molecules with height in the Atmosphere and the presence of the Earth's magnetic field, the properties of a free ionospheric plasma are extremely diverse. A large number of different waves can exist in a plasma in a magnetic field, which causes an extremely large variety of nonlinear phenomena in the ionospheric plasma (Gurevich, et al. 2007). In the case of weak nonlinearity, the main nonlinear wave process in plasma is three-wave resonance, for which the conservation laws describing such processes must be satisfied, i.e. one wave splits into two waves, two waves merge into one wave. The references also one can find in (Savin, et al. 2014).

Interesting EM wave phenomena possible connected with seismic activity and non-linear processes were observed during the RELEC Vernov experiment on December 10, 2014. Detailed analysis of these events will be presented below in subsequent sections of this article.

**2. Magnetic-wave instruments PSA – RFA as a part of the RELEC scientific payload on-board Vernov satellite.**

The RELEC scientific payload described in details by Panasyuk et al., 2016 b,c was operated on-board small spacecraft Vernov between July – December, 2014. It was named in honor of pioneer of space research Soviet academician Sergey Nikolaevich Vernov. The satellite orbit was sun -synchronous with apogee 830 km, perigee 640 km, inclination 98.4° and orbital period 100 min.

The EM emission and current in plasma in wide frequency range were measured by NChA - RChA (PSA - RFA) complex. These instruments allowed high accuracy measurements of values and fine structure of field variations.

The NChA – RChA (PSA, i.e.SAS3-R – RFA) complex of instruments consisted from the low frequency analyzer NChA (PSA – SAS3-R) and the radio frequency analyzer RChA (RFA).

The NChA (PSA – SAS3-R) instrument consisted of the following units:

- data processing unit for spectral analysis PSA;

- flux gate magnetometer, detector unit (DFM) and electronic unit (BE-FM);
- induction magnetometer IM;

- two identical complex wave probe (CWZ-1, CWZ-2).

The RChA (RFA) instrument consisted of an analyzer unit RFA-E and three-component electric antenna RFA-AE.

The NChA – RChA units were mounted on the outer 3 meters boom (IM, DFM, CWZ-1, CWZ-2, RFA-AE) and on the spacecraft thermostatic panel (units BE-FM, PSA, RFA-E). Their sizes, masses and power consumption are presented in Panasyuk et al., 2016a.

The sketch of mutual displacement of NChA at the boom is presented in Fig. 1. This instrument measured the constant magnetic field by a three-component flux gate magnetometer. The range of measured intensity was no less than  $\pm$  64000 nT, non-orthogonality of the meter component was less than 1°, the sampling frequency was 250 Hz. The mutual orthogonality of three measuring axes was provided by the DFM construction

Measurements of values and sign of three components of variable magnetic field induction vector were made with the use of the induction magnetometer IM and magnetic channels of the complex wave probes CWZ-1, CWZ-2. Each of these instruments contained one-component meter of variable magnetic field. To construct an orthogonal coordinate system mutual orthogonality of Z axes of these instruments was provided. The frequency range of the meter was from 0.1 to 40000 Hz.

Measurements of plasma current density were made by current measurement channels of the CVZ-1, CVZ-2 probes. In order to obtain correct results of measurements these probes were mounted in such a way, that their YOZ planes were parallel to the spacecraft velocity vector.

Fig. 1. Mutual orientation of PSA – SAS3-R sensors. Xsc, Ysc, Zsc – axes of the spacecraft; Vsc - velocity vector of the spacecraft; Ycwz1, Zcwz1 - measuring axes of the CWZ1; Ycwz2, Zcwz2 - measuring axes of the CWZ2.

Measurements of the potential difference were made with the use of electric channels of the CWZ-1, CWZ-2 probes, each contained the meter of potential relative the common wire.

The difference of analogue signals from CWZ-1, CWZ-2 was determined in the PSA unit. The signal from CWZ-1 relative to the common wire was also measured there. The common wire was connected with spacecraft via telemetry unit, i.e. via high resistor. Thus the potential difference between CWZ—1 place and spacecraft was determined.

The PSA unit provided:

- producing from the on-board power network  $\pm 27$  V of voltage necessary to its own operation as well as of the FM, IM, CWZ-1, CWZ-2 operation, that provided its galvanic isolation from secondary circuits;

- transmission in the digital format of FM, IM, CWZ-1, CWZ-2 outputs;

- calculation of spectral density of measurements values;

- detection of the events, i.e. unusual rare electromagnetic phenomena;

- storing measured data before and after the event;

- storage of measurement results;

- transmission of measurement results and telemetry data to the ground via spacecraft board systems.

The NChA (PSA – SAS3-R) instrument operated in three main operation modes and a command controlled recording frequency bandwidth. The main operation modes were the event

detection mode, the monitoring mode and the continuous wave recording (burst) form mode (means digitization frequency 80 kHz). The recording frequency bandwidth was from

Fig. 2. The Vernov satellite orbit, 10 December 2014, altitude ~ 670 km, local time ~ 21:10 LT, magnetic latitude ~ 750.

A sequence of unusual events was revealed on the spectral diagrams (sonograms), which were obtained for this time. It can be well traced at  $B_x$  dynamic spectrograms obtained, which

are presented in Fig. 3. The whistler with specific temporal structure can be seen in the left panel. This whistler accompanied by non-linear process in wave activity, when initial EM wave with given frequency underwent decay on two or three wave modes and then two wave structures occurred. Particular attention should be paid to the emission with increasing frequency, which was centered at ~ 5 kHz. It is presented in the right panel of Fig. 3.